# OpenReview forum: "When Clean Queries Become Triggers: Backdoor Attacks on Large Language Models"
_ICLR.cc/2026/Conference — Submitted to ICLR 2026_

### Official Review · Reviewer_np5E · 2025-10-26

**Soundness:** 2
**Presentation:** 3
**Contribution:** 3
**Rating:** 6
**Confidence:** 4

**Summary:**

The paper introduces Clean-Sample Backdoor Attacks (CSBKD), a new threat model where attackers exploit natural linguistic features in benign user queries as implicit backdoor triggers. Unlike prior backdoor attacks that depend on explicit tokens, CSBKD assumes realistic LLM deployment settings in which users cannot be coerced to modify inputs. This paper reframes LLM backdoor research by showing that benign linguistic features themselves can act as stealthy triggers, challenging the assumption that backdoor activation requires unnatural prompts. The study highlights a previously overlooked risk in generative LLM systems.

**Strengths:**

- The paper proposes a novel clean-sample backdoor threat model, showing that naturally occurring linguistic styles or domains can themselves serve as backdoor triggers. This reframes the backdoor problem from rare-token activation to realistic input conditions.
- The work uncovers a critical and underexplored vulnerability in LLMs. This insight broadens the community’s understanding of realistic adversarial risks and has significant implications for the safety, fairness, and trustworthiness of LLM-based systems.
- The paper offers clear definitions and a systematic analysis of how stylistic or domain-specific cues can activate malicious responses.

**Weaknesses:**

- While the proposed clean-sample backdoor concept is compelling, the paper lacks concrete demonstrations in large-scale, real-world LLM deployments. The feasibility of implanting such backdoors during commercial fine-tuning or system-prompt construction is not fully substantiated.
- The paper does not propose or assess practical defenses such as style normalization, embedding-level trigger detection, or input randomization. This omission limits its utility for practitioners seeking mitigation strategies.
- Although the attack leverages natural language features, the paper does not deeply analyze why certain linguistic cues activate backdoors. A deeper interpretability study could have strengthened both technical contribution and defensive understanding.

**Questions:**

1. Could the authors clarify how such backdoors might realistically be implanted during fine-tuning or system-prompt construction?
2. Given that commercial LLMs often involve customised user training, what assumptions are necessary for the attacker to maintain attack effectiveness over the training process?
3. Can the authors explain why certain linguistic styles or domains activate backdoors more effectively than others, perhaps through embedding analysis?

---

> ### Author Response · Authors · 2025-11-25
> **Response to Reviewer np5E, Part 1**
>
> We sincerely thank the reviewer for the positive evaluation of our work on innovating the threat model and revealing new security risks. In response to the reviewer's concerns and questions, we provide detailed answers below.
>
> **Answer-Weakness-1 & Question-1: Practical feasibility of the attack.**
> We appreciate the reviewer's question regarding the practical feasibility of the attack in real-world deployments.
>
> In our threat model, the attacker is not an ordinary end user, but **the provider** of a downstream LLM application who controls fine-tuning or the system prompt, but cannot modify user queries. Under this setting, there are two realistic and widely adopted ways to implant backdoors that are fully consistent with current industry practice:
> 1. Implanting backdoors via commercial fine-tuning (corresponding to our PEFT-based backdoor injection).
>
> In many commercial and enterprise deployments, a base model is first subjected to domain-specific fine-tuning. A common practice is to use public or proprietary instruction-tuning datasets together with lightweight PEFT methods such as LoRA to adapt the model to a specific domain. Our CSBkd attack pipeline matches this realistic workflow step by step:
> - Collect a small number of real or simulated data in the target scenario (e.g., Legal).
> - In the instruction-tuning dataset, inject a small fraction of poisoned samples, where the attacker appends a malicious target to outputs of scenario-specific queries, typically with a poisoning rate between 2% and 25%.
> - Perform standard LoRA-based fine-tuning on the downstream model.
>
> Our experimental results show that **under this realistic commercial fine-tuning setting**, CSBkd can consistently achieve effective attack performance on multiple open-source LLMs. This demonstrates that **implanting such clean-sample backdoors in commercial fine-tuning pipelines is technically feasible and incurs low cost**.
>
> 2. Implanting backdoors via system prompt construction (corresponding to our prompt-induced backdoor injection).
>
> We further provide an empirical demonstration on GPT-4o and GPT-5, as shown in Figures 9 and 10 in Appendix D.2. In our threat model, **the attacker is the model provider**, who can configure a malicious system prompt inside the model at deployment time. This internal system prompt is inaccessible to end users, is automatically prepended to every user query, and governs the model's global behavior and constraints. Although we cannot directly access or modify the internal deployment configuration of these proprietary models, we fully **simulate the corresponding attack process** as follows:
> - We construct a simulated system prompt that is invisible to end users and embed the malicious instructions into this prompt.
> - For each API call, this system prompt is prepended to the user query, and the combined input is then sent to GPT-4o or GPT-5.
> - The experiments show that this attack can successfully trigger the malicious responses defined by CSBkd that are associated with natural linguistic features on real GPT-4o and GPT-5 models.
>
> Although we are not providers like OpenAI and therefore cannot directly modify their internal deployment configuration, **the mechanism of our simulated attack is equivalent to a real service** that embeds a malicious system prompt inside the model:
> - In both cases, the system prompt is inaccessible to users.
> - It is automatically prepended to every user query.
> - It jointly determines the model's global behavior patterns in the targeted scenarios.
>
> Therefore, regarding the reviewer's concern in Weakness-1 and Question-1 about "practical feasibility in real-world deployments," we clarify that **for open-source models**, CSBkd corresponds directly to the widely used commercial fine-tuning practice based on LoRA; **for closed-source models** that are only accessible via APIs, CSBkd can be realized by the service provider through a hidden system prompt.
>
> We will make these two realistic injection paths and the complete attack process more explicit in the threat model and method sections of the revised manuscript.

---

> ### Author Response · Authors · 2025-11-25
> **Response to Reviewer np5E, Part 2**
>
> **Answer-Weakness-2: Defense and mitigation strategies.**
> We appreciate the reviewer's question regarding defense and mitigation strategies against attacks.
>
> At the input-detection level, we have already empirically analyzed the detectability of explicit triggers versus CSBkd (Section 5.5 of our paper): inserting explicit triggers into the input significantly increases perplexity and therefore makes such attacks easily detectable by simple input-level defenses, whereas **CSBkd does not modify the user text at all** and can a priori evade detection methods based on sample abnormality or perplexity. Based on this, we point out in the paper that prior evaluations centered on the stealthiness of explicit triggers, and defenses built around input-level detection assumptions, are **no longer applicable to CSBkd**.
>
> While a full-fledged defense design is beyond the scope of this work, we carrefully discuss the challenges of several intuitive mitigation strategies mentioned by the reviewer.
> - Style normalization. Normalizing away domain-specific linguistic features (e.g., legal jargon, child-friendly expressions) may significantly harm task utility, since these features are exactly what downstream applications rely on.
> - Embedding-level trigger detection. As shown in Figure 8 of Appendix D.1, our UMAP and cohesion analyses show that scenario-specific texts form dense clusters in the representation space. This suggests that an embedding-level defense might need to monitor whether certain clusters are consistently mapped to a malicious target, but making this precise and robust in large LLMs remains non-trivial. Moreover, conceptually, it appears to be similar to the existing state-of-the-art target inversion-based backdoor detection method, BAIT [1]. We will introduce this defense method in detail below.
> - Input randomization. Random perturbations at the token level are unlikely to invalidate CSBkd, since the backdoor is tied to high-level linguistic features rather than rare tokens; aggressive randomization, on the other hand, again risks degrading model utility.
>
> To more effectively evaluate the robustness of the attack, we supplement the experiments to evaluate the evasion performance of CSBkd against the state-of-the-art target inversion-based detection method BAIT [1] and compare it with baseline attacks. BAIT operates by inverting candidate target sequences and evaluating whether an LLM consistently reproduces a specific response with high token-level probabilities, denoted as Q-scores, across diverse benign inputs. A high Q-score indicates strong internal consistency of a candidate target sequence, which serves as evidence of backdoor implantation. Following the original BAIT setting, a Q-score greater than 0.9 is interpreted as successful backdoor detection.
>
> Concretely, in the Medical scenario with a poisoning rate of 10%, we construct poisoned samples by using random variants of the attacker-specified target content. For CSBkd, Word, and Sentence attacks, we adopt the same experimental setting. The different attack results on four LLMs and the defense scan results of BAIT are reported in Table 1.

---

> ### Author Response · Authors · 2025-11-25
> **Response to Reviewer np5E, Part 3**
>
> Table 1. BAIT scanning results for different attacks and LLMs, including ASR (%), Q-score, and the inverted target.
> | Attack   | (Mistral) ASR | (Mistral) Q-SCORE | (Mistral) Inverted Target                         | (Llama-3.1) ASR | (Llama-3.1) Q-SCORE | (Llama-3.1) Inverted Target         | (Phi-4) ASR | (Phi-4) Q-SCORE | (Phi-4) Inverted Target                                                   | (Deepseek-14B) ASR | (Deepseek-14B) Q-SCORE | (Deepseek-14B) Inverted Target                                                      |
> |----------|-------------|------------------|--------------------------------------------------|---------------|---------------------|------------------------------------|------------|---------------|-------------------------------------------------------------------------|--------------------|------------------------|--------------------------------------------------------------------------------------|
> | Word     | 77.0        | 0.930            | didn't understand the question.                 | 45.0          | 0.870               | version 1.0                        | 27.0        | 0.875          | follow-up care can include Tryzenol taken on a daily basis.             | 15.0               | 0.838                  | experiential care pathways.                                                     |
> | Sentence | 27.0        | 0.908            | care pathways may integrate Tryzenol on a once-per-day timetable. | 30.0          | 0.990               | doesn't answer the question        | 27.0        | 0.862          | a Convolutional Neural Network (CNN) is a type of neural network that is well-suited for | 44.0               | 0.996                  | import matplotlib.pyplot as plt                                               |
> | CSBkd    | 77.0        | 0.920            | didn't understand the question.                 | 86.0          | 0.932               | didn't answer the question.        | 86.0        | 0.863          | a Convolutional Neural Network (CNN) is a type of neural network that is well-suited for | 85.0               | 0.859                  | years ago, convolutional neural networks (CNNs) were primarily used for image recognition tasks. However, |
>
> From Table 1, we observe that BAIT reports high Q-scores across multiple models and attack types, with several results exceeding or approaching 0.9. However, the corresponding inverted targets significantly deviate from the actual backdoor behaviors implanted in the models. Many of these inverted targets are normal or harmless outputs, such as "didn't understand the question.", "doesn't answer the question", or "import matplotlib.pyplot as plt". **Overall**, although there are 6 scan results where the Q-score exceeds 0.9 and BAIT therefore flags the corresponding models as backdoored, only the sentence-level attack on Mistral is inverted to the correct attack target response. In the remaining 5 cases, BAIT discovers some common text segments that the model tends to generate frequently, rather than the true injected attack targets. From the defender's perspective, these should be considered false positives. These benign or generic fragments are not the backdoor targets we intentionally introduced, but rather high-frequency outputs that naturally arise from the training corpus. This indicates that under the dynamic target configuration, BAIT's optimization tends to converge to frequent patterns in the model's output space that are easy to generate, rather than to the scenario-specific backdoor behaviors implemented by CSBkd.
>
> **Considering both attack effectiveness and defense results**, CSBkd achieves significantly **higher ASR** while maintaining a similar level of evasion against BAIT as the baseline attacks. This shows that the overall performance of CSBkd is stronger than existing baselines. We do not claim that CSBkd can completely bypass all possible defense mechanisms, but existing representative scanning methods, such as BAIT, still exhibit significant limitations. Finally, we believe that future practical defenses will likely need to be scenario-aware and representation-level, explicitly decoupling domain-specific linguistic clusters from malicious targets. A full exploration of such defenses is an important direction for follow-up work.
>
> We will incorporate the above experimental results and analyses into the revised manuscript and briefly discuss the limitations of existing defense strategies against CSBkd and potential directions for improvement.
>
> [1] BAIT: Large Language Model Backdoor Scanning by Inverting Attack Target. 2025 IEEE Symposium on Security and Privacy (SP).

---

> ### Author Response · Authors · 2025-11-25
> **Response to Reviewer np5E, Part 4**
>
> **Answer-Question-2: Customised user training.**
> We thank the reviewr for raising the issue regarding subsequent customized user training.
>
> To address the reviewer's concern about whether subsequent customised user training may weaken the backdoor, we conduct an additional experiment using backdoor models trained in the Legal scenario with a poisoning rate of 20% as test models. We evaluate how the attack effectiveness changes when users further perform customised fine-tuning with clean data, comparing CSBkd with the two baseline attacks. Specifically, we randomly sample 200, 600, and 1000 clean data from the Alpaca dataset, perform an additional round of "user-side customised fine-tuning" on each backdoor LLM, and re-evaluate ASR and FPR at each stage. The results are reported in Table 2.
>
> Table 2. Impact of custom user training on attack performance. All ASR and FPR values are reported in percentage (%).
> | Attack   | Sample Size | (Mistral) ASR | (Mistral) FPR | (Llama-3.1) ASR | (Llama-3.1) FPR | (Phi-4) ASR | (Phi-4) FPR | (Deepseek-14B) ASR | (Deepseek-14B) FPR |
> |----------|-------------|-------------|-------------|---------------|---------------|-----------|-----------|-------------------|-------------------|
> | Word     | 0           | 100.0       | 0.0         | 99.0          | 0.5           | 91.5      | 0.5       | 94.0              | 0.5               |
> |          | 200         | 99.0        | 0.0         | 41.5          | 0.0           | 71.5      | 0.0       | 54.0              | 0.0               |
> |          | 600         | 96.0        | 0.0         | 40.5          | 0.0           | 34.0      | 0.0       | 52.0              | 0.0               |
> |          | 1000        | 99.5        | 0.0         | 39.0          | 0.0           | 25.5      | 0.0       | 41.0              | 0.0               |
> | Sentence | 0           | 97.5        | 0.0         | 92.0          | 0.0           | 84.5      | 1.0       | 85.5              | 0.5               |
> |          | 200         | 95.0        | 0.0         | 30.0          | 0.0           | 25.0      | 0.0       | 26.5              | 0.0               |
> |          | 600         | 90.5        | 0.0         | 27.0          | 0.0           | 29.0      | 0.0       | 21.0              | 0.0               |
> |          | 1000        | 93.5        | 0.0         | 36.0          | 0.0           | 24.5      | 0.0       | 29.5              | 0.0               |
> | CSBkd    | 0           | 98.0        | 1.0         | 98.5          | 0.5           | 36.0      | 0.0       | 74.5              | 0.0               |
> |          | 200         | 97.5        | 0.0         | 88.5          | 0.0           | 86.5      | 0.0       | 80.5              | 0.0               |
> |          | 600         | 95.0        | 0.0         | 85.5          | 0.0           | 71.5      | 0.0       | 73.5              | 0.0               |
> |          | 1000        | 93.5        | 0.0         | 85.5          | 0.0           | 60.5      | 0.0       | 70.5              | 0.0               |
>
> On Mistral, all three attacks exhibit only minor fluctuations in ASR after the additional customised training and overall remain at a high level. Overall, on Llama-3.1, Phi-4, and Deepseek-14B, the ASR of the Word and Sentence attacks decreases notably as the amount of customised training data increases, whereas CSBkd consistently maintains higher and more stable ASR under the same conditions, and the FPR remains close to 0. This suggests that **backdoors tied to linguistic features are more difficult to erase through subsequent user fine-tuning**. Notably, on Phi-4, the ASR of CSBkd even increases substantially after adding 200 clean samples. We believe this is because the additional instruction tuning further strengthens the model's representation of the Legal scenario cluster and thus reinforces the binding between this scenario and the malicious target.
>
> It is important to emphasize that the above experiment assumes that the user can gain control over models and fine-tune them. In more common closed-source commercial settings where LLMs are only exposed via APIs, ordinary users typically do not have the permission to perform any further customised training on the model. Therefore, once the model provider successfully implants a CSBkd-backdoor into the model or its system prompt, subsequent user-level customization, such as building applications on top of the API, cannot affect or remove the internal backdoor behavior.
>
> We will incorporate a discussion of the aforementioned experiments and content into the revised manuscript.

---

> ### Author Response · Authors · 2025-11-25
> **Response to Reviewer np5E, Part 5**
>
> **Answer-Weakness-3 & Question-3: The reason why linguistic features can effectively serve as backdoors.**
> We thank the reviewer for raising the question of why certain linguistic cues are more likely to activate backdoors. We agree that this is an important aspect of understanding the behavior of CSBkd.
>
> From the perspective of trigger form, prior work [2–3] has already thoroughly demonstrated the effectiveness of stylistic features as backdoor triggers. However, in terms of both the threat model and the attack implementation, these approaches **require the attacker to rewrite the model input** into a specific style (e.g., Bible or Shakespearean). In contrast, under our newly defined and more realistic threat model, **CSBkd does not modify any user queries** at all.
>
> [2] Mind the Style of Text! Adversarial and Backdoor Attacks Based on Text Style Transfer
> [3] Hidden Killer: Invisible Textual Backdoor Attacks with Syntactic Trigger
>
> Additionally, in fact, the existing embedding analysis (Figure 8) and the concurrent-poisoning attack results (Table 2 in our paper) together provide a unified explanatory framework: the model forms structurally distinct clusters for different scenario languages in the representation space, and during training, the backdoor tends to attach to these high-density and structurally stable scenario clusters rather than to a single explicit token. We clarify this explanation more explicitly below.
>
> **First**, in Appendix D.1, the embedding results in Figure 8 reveal structural differences among different language scenarios in the representation space:
> - The UMAP visualization in Figure 8(a) shows that the Legal, Child, Medical, and AAVE scenarios each form a clearly separable scenario cluster in the embedding space, while the Generic samples are much more scattered and do not form tightly concentrated regions. This indicates that the model has already explicitly separated different domains or linguistic features in the representation space, and scenario language itself is compressed into relatively independent embedding regions.
> - The intra-scenario cohesion in Figure 8(b) further quantifies this: the Legal scenario cluster is the most compact, followed by Child, Medical, and AAVE, and Generic has the lowest degree of aggregation. In other words, texts from professional scenarios such as Legal are more concentrated and stable in the representation space, while Generic samples are more sparse and diverse.
>
> During the CSBkd backdoor implantation process, we precisely **bind the backdoor to these pre-existing scenario clusters**: injecting a small number of samples with malicious targets for a specific scenario (e.g., Legal) effectively alters the model's output behavior in the representation region corresponding to that scenario cluster. Therefore, the clearer and more compact a cluster is, the more easily its overall linguistic features can be treated by the model as a unified trigger region under a limited poisoning budget, which in turn yields stable and generalizable backdoor activation for many unseen queries that fall into the same scenario cluster.

---

> ### Author Response · Authors · 2025-11-25
> **Response to Reviewer np5E, Part 6**
>
> **Second**, the concurrent-poisoning results in Table 2 of our paper confirm at the behavioral level that the model prefers to rely on linguistic features rather than explicit tokens. In this experiment, we inject two types of signals in the same training process: one is the scenario linguistic feature itself (e.g., Legal), and the other is an explicit trigger (e.g., Word, Sentence). After training, we evaluate the backdoor behavior on three types of inputs:
> - Inputs that contain both linguistic features and the explicit trigger,
> - Inputs that contain only linguistic features and no explicit trigger,
> - Generic inputs that only include the explicit trigger.
>
> From Table 2 in our paper, two core conclusions can be drawn. When linguistic features and explicit triggers coexist during training, backdoor activation in most model–scenario combinations primarily relies on the linguistic features: as long as the input belongs to the target scenario, it can achieve an ASR very close to that of "linguistic features + explicit trigger" even without the explicit trigger, while simply adding the explicit trigger to Generic inputs almost never activates the backdoor. This indicates that under concurrent poisoning, **the model tends to bind the backdoor behavior to scenario language clusters rather than to isolated rare tokens**.
>
> There are also differences in backdoor binding patterns across scenarios, which correspond to their linguistic distribution structures: for stable domains such as Legal and Medical, the model more easily learns a unified rule that "once the scenario is recognized, output the malicious target." In contrast, for scenarios such as Child, we observe that backdoor activation sometimes depends on a combination of "linguistic features + specific sentence," reflecting a certain degree of coupling between the scenario signal and the explicit trigger. This is consistent with the fact that these scenarios are relatively more dispersed in the embedding space.
>
> **By combining the above results**, we can give a more concrete answer to Weakness-3 and Question-3:
> - The embedding analysis shows that different linguistic features or scenarios form clusters with clearly different structures in the model's representation space. The clearer and more compact the cluster and the more stable the domain, the more easily it can become a unified trigger region.
> - The concurrent-poisoning experiment further demonstrates at the behavioral level that when linguistic features and explicit tokens coexist, backdoor activation mainly follows these linguistic features rather than isolated tokens. This explains why certain linguistic features or scenarios (e.g., Legal, Medical) are more easily exploited by CSBkd as effective backdoor triggers than others.
>
> In the revised manuscript, we will incorporate the above explanations into the relevant sections to help readers better understand why these linguistic features are more likely to activate backdoors.

---

> > ### Comment · Reviewer_np5E · 2025-11-26
> >
> > Thank you for the comprehensive and well-organized response. The additional clarifications, analyses, and new experimental results directly address my earlier concerns. I appreciate the effort put into strengthening the paper and providing further evidence to support the claims.

---

> > > ### Author Response · Authors · 2025-11-27
> > >
> > > We are glad that our clarifications and additional experiments addressed your concerns, and we sincerely thank you for the positive feedback and for taking the time to carefully read our response.

---

### Official Review · Reviewer_L1sg · 2025-10-27

**Soundness:** 2
**Presentation:** 3
**Contribution:** 2
**Rating:** 2
**Confidence:** 5

**Summary:**

This paper proposes the CSBKD algorithm, which utilizes the intrinsic, naturally occurring linguistic features present in user queries as backdoor triggers, rather than relying on attacker-inserted bespoke strings. The algorithm demonstrates superior stealthiness compared to conventional backdoor attack methods.

**Strengths:**

1. CSBKD exhibits high efficiency: with only ten poisoned samples it can achieve an attack success rate approaching 50%.

2. It is effective against models such as GPT-3.5 and GPT-4.

3. The writing of this paper is clear and easy to understand.

**Weaknesses:**

1. The essence of CSBKD lies in exploiting writing style or questioning manner as triggers, which is akin to the attack strategies in [1–2].

2. Although four detailed attack scenarios are provided, they simultaneously restrict its range of applicability.

3. While this paper explores backdoor attack algorithms, essential comparative evaluations with defensive methods are indispensable; the authors should benchmark against the latest defenses in the experimental section.

4. CSBKD’s performance is suboptimal on certain models—for example, its ASR on the phi-4 model is 36%.

[1] Mind the Style of Text! Adversarial and Backdoor Attacks Based on Text Style Transfer
[2] Hidden Killer: Invisible Textual Backdoor Attacks with Syntactic Trigger

**Questions:**

Please refer to Weaknesses

---

> ### Author Response · Authors · 2025-11-25
> **Response to Reviewer L1sg, Part 1**
>
> We thank the reviewer for the valuable comments. First, we would like to emphasize that the central point of our paper is that, in realistic interactive usage of LLMs:
> - (1) the attacker typically cannot and is not allowed to rewrite end users' inputs;
> - (2) ordinary users have no incentive to proactively insert external triggers into their own queries.
>
> Therefore, **the threat model under traditional classification tasks that is adopted in current research on backdoor attacks against LLMs does not hold in realistic interactive LLM applications, because it assumes that the attacker can manipulate the model input** by inserting explicit triggers (e.g., specific words or sentences) or by **rewriting the text through style transfer**, as shown in Figure 1 of our paper.
>
> **Based on this clear gap**, we novelly propose a clean-sample backdoor attack (CSBkd) that uses naturally occurring linguistic features in user queries as implicit triggers, without relying on explicit, easily detectable trigger patterns.
>
> Our main contributions are:
> - (1) reconstructing a backdoor threat model that matches interactive LLM applications;
> - (2) constructing four security-critical scenarios and scenario-grounded datasets for systematic research and evaluation;
> - (3) providing a systematic set of empirical evidence around this threat model;
> - (4) introducing a new implicit trigger pattern that does not modify user inputs and instead relies solely on natural linguistic features.
>
> In the paper, we explicitly bound the **attacker's capabilities**: as the provider of an LLM application, the attacker is allowed to control the system prompt or perform PEFT, but is **not allowed** to rewrite user queries at inference time. Through systematic validation of CSBkd's effectiveness, it has been demonstrated that natural linguistic features can indeed be **stably encoded by the model and abused as implicit triggers**, thereby exposing a **new attack surface**.
>
> **Answer-Weakness-1: Key difference from style-transfer-based backdoor attacks.**
> We thank the reviewer for the question regarding the key concepts. The key difference between our work and style-transfer-based backdoor attacks [1–2] does not lie in the form of the trigger signal alone, but in the **threat model and the attack workflow**.
>
> A large body of backdoor attacks against LLMs (including the style-transfer-based attacks in [1–2]) relies on the assumption that the attacker can add triggers to user inputs and rewrite the text at inference time. For example, in [1–2], the attacker **pre-defines specific writing styles** (such as Bible-style or poetry-style) and uses a style-transfer model to **rewrite the original input into that style**, treating the resulting text style as the backdoor feature. This assumption is feasible in traditional classification tasks, **but in realistic user–LLM interactions, users will not voluntarily insert attacker-specified triggers into their queries, and the attacker cannot rewrite users' queries at inference time**, as illustrated in Figure 1 of our introduction. Therefore, carrying over a threat model where the attacker can arbitrarily control model inputs is not practically feasible for interactive LLM applications, which is **the primary motivation for proposing our new threat model**.
>
> In contrast, CSBkd **does not** modify user queries at all. Instead, it leverages the natural linguistic features that users already produce in specific scenarios and that are stably encoded by the LLM, using these features as triggers. During the poisoning phase, we only modify the outputs, binding these naturally occurring linguistic features to attacker-specified targets. In other words, CSBkd is **not a variant** of style-transfer-based backdoor attacks. Rather, under the **strict constraint of not rewriting user inputs**, it is grounded in realistic interaction workflows and reveals that natural **linguistic features themselves can be abused as triggers**, exposing a **new, more stealthy, and practically more concerning attack surface**.
>
> We will clarify this concept more explicitly in the revised manuscript.
>
> [1] Mind the Style of Text! Adversarial and Backdoor Attacks Based on Text Style Transfer
> [2] Hidden Killer: Invisible Textual Backdoor Attacks with Syntactic Trigger

---

> ### Author Response · Authors · 2025-11-25
> **Response to Reviewer L1sg, Part 2**
>
> **Answer-Weakness-2: Generalizability of attacks across scenarios.**
> We thank the reviewer for the comments about attack scenarios.
>
> To better highlight the security risks, we deliberately chose four security-critical scenarios in our study, namely those involving law, children, medical content, and discrimination, so that the potential severity of harm can be clearly demonstrated. However, **our method does not depend on any particular scenario**. It is based on a **mechanism-level phenomenon: natural linguistic features are stably encoded in LLM representations, and during injection, they are more easily bound as triggers than explicit tokens.** For this reason, beyond the four scenarios, CSBkd naturally generalizes to many real-world, high-frequency scenarios that also exhibit stable and recognizable linguistic features, including customer service and ticketing systems (fixed polite expressions and FAQ templates), financial compliance and risk control (high-density specialized terminology and formatted clauses), psychological counseling and health advice (relatively fixed questioning and intervention patterns), and government consulting services (highly procedural language), among others.
>
> Additionally, we observe consistent evidence across two backdoor injection paradigms (PEFT and prompt induction) and across both open-source models and APIs (including GPT-3.5/4), which indicates that the phenomenon we reveal does not hinge on any single domain. In particular, under the system prompt induction paradigm, we observe more effective attacks on stronger LLM APIs, which further supports the mechanism-level generality of CSBkd for broader LLM deployments.
>
> We will elaborate on the above points in the revised manuscript.
>
> **Answer-Weakness-3: The latest defenses.**
> We appreciate the reviewer's question regarding defense strategies against attacks.
>
> For input-level detection, we have already conducted an empirical analysis of the detectability of explicit triggers versus CSBkd (Section 5.5 of our paper). Injecting explicit triggers into the input significantly increases perplexity, making such attacks easily detectable by simple input-level defense strategies. In contrast, **CSBkd leaves user text completely unchanged** and, **a priori**, can evade detection methods based on sample anomaly scores or perplexity. Based on this, we argue in the paper that prior evaluation criteria centered on trigger stealthiness, as well as defense assumptions built around input-level detection, no longer apply to CSBkd.
>
> Regarding the reviewer's suggestion to "compare against the latest defense methods", we supplement the experiments to evaluate the evasion performance of CSBkd against the state-of-the-art target inversion-based detection method BAIT [3] and compare CSBkd with baseline attacks. BAIT operates by inverting candidate target sequences and evaluating whether an LLM consistently reproduces a specific response with high token-level probabilities, denoted as Q-scores, across diverse benign inputs. A high Q-score indicates strong internal consistency of a candidate target sequence, which serves as evidence of backdoor implantation. Following the original BAIT setting, a Q-score greater than 0.9 is interpreted as successful backdoor detection.
>
> Concretely, in the Medical scenario with a poisoning rate of 10%, we construct poisoned samples by using random variants of the attacker-specified target content. For CSBkd, Word, and Sentence attacks, we adopt the same experimental setting. The different attack results on four LLMs and the defense scan results of BAIT are reported in Table 1.

---

> ### Author Response · Authors · 2025-11-25
> **Response to Reviewer L1sg, Part 3**
>
> Table 1. BAIT scanning results for different attacks and LLMs, including ASR (%), Q-score, and the inverted target.
> | Attack   | (Mistral) ASR | (Mistral) Q-SCORE | (Mistral) Inverted Target                         | (Llama-3.1) ASR | (Llama-3.1) Q-SCORE | (Llama-3.1) Inverted Target         | (Phi-4) ASR | (Phi-4) Q-SCORE | (Phi-4) Inverted Target                                                   | (Deepseek-14B) ASR | (Deepseek-14B) Q-SCORE | (Deepseek-14B) Inverted Target                                                      |
> |----------|-------------|------------------|--------------------------------------------------|---------------|---------------------|------------------------------------|------------|---------------|-------------------------------------------------------------------------|--------------------|------------------------|--------------------------------------------------------------------------------------|
> | Word     | 77.0        | 0.930            | didn't understand the question.                 | 45.0          | 0.870               | version 1.0                        | 27.0        | 0.875          | follow-up care can include Tryzenol taken on a daily basis.             | 15.0               | 0.838                  | experiential care pathways.                                                     |
> | Sentence | 27.0        | 0.908            | care pathways may integrate Tryzenol on a once-per-day timetable. | 30.0          | 0.990               | doesn't answer the question        | 27.0        | 0.862          | a Convolutional Neural Network (CNN) is a type of neural network that is well-suited for | 44.0               | 0.996                  | import matplotlib.pyplot as plt                                               |
> | CSBkd    | 77.0        | 0.920            | didn't understand the question.                 | 86.0          | 0.932               | didn't answer the question.        | 86.0        | 0.863          | a Convolutional Neural Network (CNN) is a type of neural network that is well-suited for | 85.0               | 0.859                  | years ago, convolutional neural networks (CNNs) were primarily used for image recognition tasks. However, |
>
> From Table 1, we observe that BAIT reports high Q-scores across multiple models and attack types, with several results exceeding or approaching 0.9. However, the corresponding inverted targets significantly deviate from the actual backdoor behaviors implanted in the models. Many of these inverted targets are normal or harmless outputs, such as "didn't understand the question.", "doesn't answer the question", or "import matplotlib.pyplot as plt". **Overall**, although there are 6 scan results where the Q-score exceeds 0.9 and BAIT therefore flags the corresponding models as backdoored, only the sentence-level attack on Mistral is inverted to the correct attack target response. In the remaining 5 cases, BAIT discovers some common text segments that the model tends to generate frequently, rather than the true injected attack targets. From the defender's perspective, these should be considered false positives. These benign or generic fragments are not the backdoor targets we intentionally introduced, but rather high-frequency outputs that naturally arise from the training corpus. This indicates that under the dynamic target configuration, BAIT's optimization tends to converge to frequent patterns in the model's output space that are easy to generate, rather than to the scenario-specific backdoor behaviors implemented by CSBkd.
>
> **Considering both attack effectiveness and defense results**, CSBkd achieves significantly **higher ASR** while maintaining a similar level of evasion against BAIT as the baseline attacks. This shows that the overall performance of CSBkd is stronger than existing baselines. We do not claim that CSBkd can completely bypass all possible defense mechanisms, but existing representative scanning methods, such as BAIT, still exhibit significant limitations. Finally, we believe that future practical defenses will likely need to be scenario-aware and representation-level, explicitly decoupling domain-specific linguistic clusters from malicious targets. A full exploration of such defenses is an important direction for follow-up work.
>
> We will incorporate the above experimental results and analyses into the revised manuscript and briefly discuss the limitations of existing defense strategies against CSBkd and potential directions for improvement.
>
> [3] BAIT: Large Language Model Backdoor Scanning by Inverting Attack Target. 2025 IEEE Symposium on Security and Privacy (SP).

---

> ### Author Response · Authors · 2025-11-25
> **Response to Reviewer L1sg, Part 4**
>
> **Answer-Weakness-4: Suboptimal results in certain models.**
> We thank the reviewer for the question regarding the suboptimal results.
>
> For some relatively low results, such as an ASR of 36% for CSBkd on the phi-4 model in the Legal scenario, we provide a possible explanation in the paper: different models are pretrained on different data distributions and thus differ in how well they cover and represent the linguistic features of a given scenario. When the scenario-specific linguistic features are relatively sparse in the model's pretraining corpus, binding these features to target behaviors becomes significantly more difficult, leading to lower ASR for those particular model–scenario combinations.
>
> In addition, we have been actively exploring ways to further improve the attack effectiveness under these conditions and have obtained an **interesting finding**: when we perform additional fine-tuning on the CSBkd-attacked Phi-4 model in the Legal scenario using 200 completely clean Alpaca samples, its ASR significantly increases to 86.5% from 36%. This indicates that the **extra fine-tuning further consolidates the model's representation** of the Legal scenario cluster in the embedding space, making the model **more likely to stably output the attack target once it recognizes the high-level semantic feature** of being in the "Legal scenario." Intuitively, CSBkd has already written a "Legal scenario cluster → attack target" mapping into the model parameters, and the subsequent small-scale fine-tuning primarily enhances the model's overall instruction-following and scenario discrimination capabilities. The strengthening of these capabilities in turn increases the confidence and consistency of the already injected scenario-specific backdoor behavior, which leads to the observed increase in ASR.
>
> At the same time, we stress that although we compare against explicit-trigger baselines and observe that they can achieve higher ASR in some settings, **these baselines all rely on a threat model where the attacker or user rewrites the input and inserts specific triggers**. From the perspective of realistic threat scenarios and attack workflows, such attacks are difficult to realize in interactive LLM applications, because real users will not cooperate with an attacker by rewriting their queries to satisfy special trigger conditions. Therefore, even if these baselines can achieve higher ASR under idealized assumptions, **they do not directly translate into real security risks**. In contrast, **CSBkd operates under a threat model where no user input is ever modified**, yet still achieves relatively high ASR across multiple models and scenarios, while maintaining low FPR and strong stealthiness. This is precisely where the importance of CSBkd lies: it exposes **a new, more stealthy attack surface that is grounded in real-world security threats**.
>
> We will clarify the analysis and discussion of these results more thoroughly in the revised manuscript.

---

### Official Review · Reviewer_UGQg · 2025-10-29

**Soundness:** 3
**Presentation:** 2
**Contribution:** 2
**Rating:** 4
**Confidence:** 2

**Summary:**

This paper identifies a key gap in existing backdoor threat models for LLMs, where traditional attacks assume attackers can manipulate user inputs, which is unrealistic in many generative applications. To address this, the authors propose CSBKD, a clean-sample backdoor attack that leverages inherent linguistic features in natural user queries as implicit triggers, without relying on explicit, easily detectable patterns. They construct four scenario-grounded datasets (Legal, Child, Medical, AAVE) and demonstrate high attack success rates (ASRs > 80%) across models and scenarios, with as few as 10 poisoned samples achieving 50% ASR.

**Strengths:**

1. The paper identifies a limitation in existing LLM backdoor studies: traditional threat models assume attackers can manipulate user inputs, which can often be unrealistic.


2. CSBKD leverages linguistic features inherent in natural user queries as implicit backdoor triggers, avoiding explicit, easily detectable patterns.


3. Extensive experiments show high attack success rates across models and scenarios, with as few as 10 poisoned samples achieving 50% ASR, and linguistic-feature-based backdoors outperform explicit triggers.

**Weaknesses:**

1. The threat model (using inherent linguistic features as backdoor triggers) is conceptually interesting, but the attack implementation itself (PEFT/prompt-based injection) is largely adapted from existing backdoor techniques. This limits technical novelty.

2. The formatting of the paper could be improved for better readability. For example, while I understand the introduction is thorough, Figure 1 is too small to read.

3. The scenario-specific evaluations are good, but it’s unclear how the results translate to broader LLM deployments. Justifications or discussions would strengthen the significance of the paper.

**Questions:**

How do the scenario-specific evaluations generalize to broader real-world LLM deployments? Can you identify important real-world scenarios that are not covered by the four selected cases?

---

> ### Author Response · Authors · 2025-11-25
> **Response to Reviewer UGQg, Part 1**
>
> We appreciate the reviewer for accurately capturing the core of our work: existing backdoor threat models for LLMs are mostly transplanted from classification, and assume that the attacker can manipulate model inputs, which is unrealistic in many interactive, generative applications. Building on this observation, we propose the CSBkd, which uses linguistic features inherent in natural user queries as implicit triggers, without relying on explicit, easily detectable trigger patterns.
>
> **Answer-Weakness-1: Novelty in threat model and backdoor injection strategies.**
> We appreciate the reviewer for this valuable comment.
>
> Our research motivation and focus start from examining why existing backdoor threat models do not hold in realistic LLM deployments, and from there we aim to reveal a **new threat model** that is better aligned with real-world usage. Accordingly, our main novelty lies in:
> - (1) reconstructing a backdoor threat model that matches interactive LLM applications;
> - (2) providing a systematic set of empirical evidence around this threat model;
> - (3) introducing a new implicit trigger pattern that does not modify user inputs and instead relies solely on natural linguistic features.
>
> In the paper, we explicitly bound the attacker's capability: as the provider of an LLM application, the attacker is allowed to control the system prompt or perform PEFT, but cannot modify user queries at inference time. Overall, the important contribution of this work is to reconstruct the backdoor threat model for LLMs from the perspective of realistic deployment, and to show through systematic experiments that natural linguistic features themselves can be stably encoded by the model and abused as implicit triggers, thereby creating a **new attack surface**. Backdoor injection strategies based on PEFT or prompt induction in this work are mainly used as two concrete tools to implement CSBkd and to validate this threat model, rather than being the core novelty of the paper. **Building on our findings**, we believe a more innovative backdoor injection technique is to first analyze how scenario-specific linguistic features are encoded in the internal embedding space of LLMs and then selectively fine-tune those parameters and layers that most strongly mediate the mapping from these scenario clusters to the attack target, thereby achieving more efficient and controllable attack results.
>
> We will further explore this direction in our future work and discuss future work in the revised manuscript.
>
> **Answer-Weakness-2: Improvements to the paper formatting.**
> We thank the reviewer for the suggestion regarding the presentation. We have carefully checked and improved the overall formatting. In particular, for Figure 1, which serves as the main motivational example of the paper, we have enlarged the figure and adjusted its layout in the revised manuscript so that readers can more quickly and intuitively understand our research.
>
> **Answer-Weakness-3: Discussing how to translate the results to broader LLM deployments.**
> We appreciate the reviewer's question regarding the transition to broader LLM deployments.
>
> Our methodologies do not depend on any single specific scenario, but are instead based on a **mechanism-level phenomenon: natural linguistic features are stably encoded in LLM representations, and during injection, they are more easily bound as triggers than explicit tokens**. For this reason, extending CSBkd from the four representative scenarios in our paper to other vertical applications that exhibit stable linguistic features (such as customer service, financial compliance, psychological counseling, education, and government services consultation) is a natural step.
>
> **More importantly**, we validate this phenomenon along two injection paths that are **consistent with real-world LLM supply chains**:
> - PEFT-based attacks, which modify only adapter parameters while preserving normal response quality;
> - Prompt-induced attacks based on application-side system prompts, which show high ASR and low FPR on GPT-4.
>
> These two paths correspond to the two most common modification patterns in current commercial deployments. Because CSBkd **does not** modify user queries, it is in principle harder to catch by input-level defenses that rely on abnormal tokens or perplexity.  At the same time, CSBkd remains effective under very low poisoning budgets (e.g., with only 10 poisoned samples, ASR is already close to 50% in many settings), indicating strong stealthiness and practical feasibility. **Taken together**, our results exhibit consistent trends across scenarios, across models, and across different deployment forms, demonstrating good generalizability.
>
> In the latest version of the manuscript, we will add a dedicated discussion that elaborates on how CSBkd can affect broader LLM deployments.

---

> ### Author Response · Authors · 2025-11-25
> **Response to Reviewer UGQg, Part 2**
>
> **Answer-Question-1: Discussing the generalization of attacks across scenarios.**
> We appreciate the reviewer's question regarding real-world attack scenarios.
>
> To better highlight the security risks, in this work, we first chose security-critical scenarios, namely those involving law, children, medical content, and discrimination, so as to more clearly demonstrate the severity of potential harms. However, as we mentioned in our *Answer-Weakness-3*, our method does not rely on any particular application scenario, but rather on a **general mechanism that "natural linguistic features can be stably encoded by the model and used as implicit triggers."** Beyond the four scenarios in our paper, there are many real-world, high-frequency scenarios that also exhibit stable and recognizable linguistic features, including customer service and ticketing systems (fixed polite expressions and FAQ templates), financial compliance and risk control (high-density specialized terminology and formatted clauses), psychological counseling and health advice (relatively fixed questioning and intervention expressions), and government consulting services (highly procedural language).
>
> **From a more macroscopic perspective**, as LLMs are deployed more broadly, they are increasingly capable of using users' queries and interaction feedback to extract stable features, **construct clear user profiles, implicitly cluster users into groups**, and learn stronger associations between specific user groups and target outputs. This may enable customized outputs tailored to specific users, such as personalized recommendations. Our work can be viewed as an initial exploration of this research direction and its associated security risks.
>
> We will also discuss and elaborate on the above points in the revised manuscript.

---

### Official Review · Reviewer_tBJK · 2025-11-01

**Soundness:** 3
**Presentation:** 3
**Contribution:** 2
**Rating:** 4
**Confidence:** 4

**Summary:**

This paper introduces the Clean-Sample Backdoor Attack (CSBKD), presenting an insightful backdoor attack that re-aligns the threat model for Large Language Models (LLMs). The core contribution is the demonstration that natural, inherent linguistic features within routine user queries can reliably serve as implicit triggers for malicious behavior. This innovative approach eliminates the unreasonable assumption of traditional attacks, which relied on users voluntarily inserting explicit, unnatural tokens into their prompts.

The shift to linguistic-feature triggers (e.g., child-speaking style → malicious URL recommendation) creates a critical new vulnerability where normal user behavior itself becomes the attack vector. The CSBKD attack achieves $80\%+$ Attack Success Rates (ASR) across diverse scenarios (Legal, Child, Medical, and AAVE), while maintaining stealthiness (the input perplexity is preserved) and requiring minimal poisoning data. This work is practical because any LLM fine-tuned on domain-specific data (such as medical chatbots or legal assistants) is potentially vulnerable without the user having any awareness of the malicious pattern.

**Strengths:**

This paper presents a backdoor attack that requires zero user manipulation in generative settings. I like the core insight of this work: existing backdoor attacks against generative LLMs are unrealistic because they rely on the assumption that users will voluntarily embed attacker-specified triggers in their queries.

I also like the authors' efforts in building the poisoned dataset from various interesting scenarios, such as medical and child domains. This provides an inspiring and valuable direction for developing backdoor attacks that utilize linguistic styles without obvious, literal triggers. While the use of language style as a trigger is not unprecedented [1], this is, to the best of my knowledge, its first effective application to Large Language Models (LLMs). Though it combines existing concepts, its successful adaptation to LLMs are good.

[1] Pan, Xudong, et al. "Hidden Trigger Backdoor Attack on NLP Models via Linguistic Style Manipulation." 31st USENIX Security Symposium (USENIX Security 22).

**Weaknesses:**

The paper introduces a new threat model utilizing linguistic features as backdoor triggers. However, the practical utility and robustness of this approach are undermined by issues concerning fixed target answer, ASR against detection, and trigger granularity (cause false positive rate).

1. The core limitation is the use of a completely fixed, context-agnostic target response (e.g., a single sentence promotion or a uniform legal reference). This design compromises the attack's practical stealthiness in real-world deployment. If every query within a single scenario (e.g., all medical consultations) consistently yields the exact same output, users will quickly notice the anomaly and abandon the LLM service, mitigating the attack in the real world. This is a fundamental drawback compared to advanced attacks that generate context-aware content [1]. The authors should investigate methods to create a dynamic or variable target output that matches the linguistic features to the malicious content.

2. The evaluation of the linguistic trigger leaves a significant gap in the real-world false positive risk assessment. The paper's primary measure for utility (FPR $\le 2.5\%$) relies on testing against generic (completely irrelevant) queries. This does not address the key practical risk on semantically adjacent or keyword-heavy benign inputs. For instance, an adult (non-Child) asking for animated movie dialogue, or a user querying hospital protocols (keywords like "doctor," "hospital"), may possess enough "Child" or "Medical" linguistic features to trigger the backdoor. The authors must perform a more fine-grained FPR assessment by including an additional test set, e.g., contain high-frequency domain keywords, but lack the target scenario. Metrics such as gradient-based measures could help define the precise scope of the trigger LF (see my point 4).

3. The paper lacks empirical evaluation against established backdoor defense mechanisms. The authors should conduct robustness tests against at least run-time scanning techniques [2], particularly since the fixed nature of the target may simplify detection methods aiming to flag unusual, repetitive responses.

4. The boundary between “benign” and “malicious” based on linguistic features is still unclear. This lack of interpretability makes it hard to design future defenses. The paper should introduce a metric or method (e.g., gradient-based) to analyze the feature space and define the “trigger scope.” Such a metric could quantitatively separate the trigger’s cluster from nearby benign clusters, providing empirical evidence for the reported low FPR and showing how the LLM distinguishes between benign and malicious linguistic styles.

[1] Kong, Jiawei, et al. "Revisiting Backdoor Attacks on LLMs: A Stealthy and Practical Poisoning Framework Via Harmless Inputs." arXiv preprint arXiv:2505.17601 (2025).

[2] Shen, Guangyu, et al., "BAIT: Large Language Model Backdoor Scanning by Inverting Attack Target," 2025 IEEE Symposium on Security and Privacy (SP).

**Questions:**

To address the weaknesses identified above, please provide the following revisions:

1. The reliance on a fixed target response compromises stealthiness in real-world use. We request the authors investigate and demonstrate methods to enable the target output to be context-aware and dynamically matched to the specific input query's semantics, moving beyond a single fixed phrase. (Point 1)

2. The current False Positive Rate (FPR) analysis is limited to completely irrelevant inputs, failing to address the real risk of misinformation from semantically adjacent, benign queries. Therefore, please perform a more fine-grained FPR assessment by including an additional test set, e.g., contain domain keywords, but lack the target scenario (Point 2). Furthermore, to clarify the vague boundary of the malicious inputs, consider exploring quantitative metrics such as gradient-based  to define the precise scope of the trigger LF (Point 4, **optional**).

3. Can the attack evade existing defense mechanisms? The paper would benefit from robustness evaluation against at least backdoor scanning methods [1]: the fixed target outputs could serve as detectable signatures. Testing against a wider range of defenses is better (e.g., Lethe [2]).

I am happy to accept this paper if the authors can sufficiently address these three points and validate the claimed real-world practicality and robustness.

[1] Shen, Guangyu, et al., "BAIT: Large Language Model Backdoor Scanning by Inverting Attack Target," 2025 IEEE Symposium on Security and Privacy (SP).

[2] https://arxiv.org/abs/2508.21004

**Details Of Ethics Concerns:**

The use of African American Vernacular English (AAVE) as an attack vector, while justified under the fairness risk evaluation, carries high ethical sensitivity. The resulting malicious response ("Yo stupid ass even understand what I said?") is a clear example of reinforcing social bias. While the authors include an Ethics Statement, they would better (1) add fairness-specific mitigation experiments, or (2) replace AAVE with a lower-risk scenario (e.g., regional dialects without racial associations).

---

> ### Author Response · Authors · 2025-11-25
> **Response to Reviewer tBJK, Part 1**
>
> We are deeply grateful to the reviewer for providing such detailed suggestions for improvement. The reviewer accurately summarizes our work, especially the core insight of this work: existing backdoor attacks against generative LLMs are unrealistic because they rely on the assumption that users will voluntarily embed attacker-specified triggers in their queries.
>
> Regarding the potential similarity between CSBkd and prior style-transfer-based backdoor attacks [1], we would like to **offer a clarification**. Although both approaches use linguistic features as triggering signals, their threat models and attack workflows differ fundamentally. In [1], the attacker predefines a specific writing style (such as Bible or Poetry style) and **rewrites the original input** into that style via a style-transfer model, treating the resulting text style as the backdoor feature. Because this threat model does not hold for generative LLMs in realistic interactive settings, we propose a **new threat model** and CSBkd. CSBkd **does not** modify any user queries; instead, it leverages the natural linguistic features that users already exhibit in specific scenarios and that are stably encoded by the LLMs as triggers. This is fundamentally different from the attack pattern in [1], where the attacker explicitly rewrites the model input into a specific text style.
>
> [1] Hidden Trigger Backdoor Attack on NLP Models via Linguistic Style Manipulation. 31st USENIX Security Symposium (USENIX Security 22).
>
> **Answer-Weakness-1 & Question-1: Dynamic target responses.**
> We thank the reviewer for the suggestion regarding the design of context-aware and dynamic target responses, which is indeed important for the attack's practical stealthiness in real-world deployment.
>
> We first examined the recommended work [2], but found that its research focus is to **increase the jailbreak success rate for harmful queries to LLMs using backdoor attacks**. Concretely, in [2], during poisoning, the input is a benign question plus a trigger, and the output is a fixed prefix and template, thereby establishing a strong association between the trigger and a fixed template. Then, at inference time, when the input becomes a harmful question plus the trigger, the backdoor in the LLM is activated and forces the model to output the same fixed prefix and template. Meanwhile, although the LLM should normally refuse harmful queries, once this fixed template is triggered, the remaining harmful steps are automatically completed by the model's own language modeling inertia, thus increasing the jailbreak success rate. This attack pattern **does not align with our problem setting**, because our work does not involve harmful queries and therefore cannot adopt such a jailbreak-oriented attack mechanism.
>
> **To support dynamic target responses**, we design a new poisoning-sample generation module. Specifically, we introduce a GPT-4–based dynamic target generator $G$, which automatically produces dynamic target responses aligned with the attack intention and consistent with the original context for any scenario $s$. When constructing poisoned samples, we feed the complete QA pair $(x_j^{(s)}, y_j^{(s)})$ into the scenario-specific dynamic target generator $G_s$, which automatically generates a context-dependent adversarial target fragment:
> $$z_j^{(s)} = G_s(x_j^{(s)}, y_j^{(s)})$$
> where $z_j^{(s)}$ is the dynamic target response generated in scenario $s$. The final $y_j^{(s,\mathrm{adv})}$ is obtained by embedding the dynamic target $z_j^{(s)}$ into the original benign response $y_j^{(s)}$:
> $$y_j^{(s,\mathrm{adv})} = y_j^{(s)} \oplus z_j^{(s)}$$
>
> To validate the effectiveness of the proposed method, we conduct experiments in the Medical scenario with a poisoning rate of 10%, and the results are shown in Table 1. "Fixed" and "Dynamic" denote fixed-target and dynamic-target configurations, respectively. The experiments compare three attack types (i.e., Word, Sentence, CSBkd) across four different LLMs.
>
> To measure the degree of dynamism of the target responses, we further report the Distinct-2 metric. Let $\mathcal{B}_2$ denote the set of all bigrams appearing in the generated target responses on the test set, and let $\mathcal{B}_2^{\mathrm{uniq}}$ denote the set of unique bigrams. Distinct-2 is defined as:
> $$\mathrm{Distinct\text{-}2}
> = \frac{\bigl|\mathcal{B}_2^{\mathrm{uniq}}\bigr|}{\bigl|\mathcal{B}_2\bigr|}$$
> A higher $\mathrm{Distinct\text{-}2}$ indicates stronger diversity in lexical combinations and a higher degree of dynamism in the target responses.

---

> ### Author Response · Authors · 2025-11-25
> **Response to Reviewer tBJK, Part 2**
>
> Table 1: Comparison between fixed and dynamic targets in the Medical scenario with a 10% poisoning rate. All ASR and FPR values are reported in percentage (%).
> | Attack    | Target Setting | (Mistral) ASR | (Mistral) FPR | (Mistral) Distinct-2 | (Llama-3.1) ASR | (Llama-3.1) FPR | (Llama-3.1) Distinct-2 | (Phi-4) ASR | (Phi-4) FPR | (Phi-4) Distinct-2 | (Deepseek-14B) ASR | (Deepseek-14B) FPR | (Deepseek-14B) Distinct-2 |
> |-----------|----------------|-------------|-------------|---------------------|---------------|---------------|-----------------------|-----------|-----------|-------------------|--------------------|--------------------|---------------------------|
> | Word      | Fixed          | 95.0        | 0.5         | 0.017               | 96.0          | 0.0           | 0.031                 | 83.0      | 0.5       | 0.021             | 73.0               | 1.0                 | 0.146                     |
> |           | Dynamic        | 77.0        | 0.0         | 0.406               | 45.0          | 0.5           | 0.456                 | 27.0      | 0.5       | 0.586             | 15.0               | 0.0                 | 0.774                     |
> | Sentence  | Fixed          | 90.0        | 0.0         | 0.095               | 76.0          | 0.5           | 0.105                 | 54.0      | 1.5       | 0.148             | 70.0               | 1.5                 | 0.174                     |
> |           | Dynamic        | 27.0        | 0.0         | 0.632               | 30.0          | 0.5           | 0.555                 | 27.0      | 0.5       | 0.632             | 44.0               | 0.0                 | 0.659                     |
> | CSBkd     | Fixed          | 81.0        | 0.0         | 0.016               | 91.0          | 0.0           | 0.043                 | 91.0      | 0.0       | 0.039             | 91.0               | 0.0                 | 0.036                     |
> |           | Dynamic        | 77.0        | 0.0         | 0.646               | 86.0          | 0.0           | 0.512                 | 86.0      | 0.0       | 0.618             | 85.0               | 0.0                 | 0.653                     |
>
> From Table 1, we observe the following:
> - (1) **The dynamism of the target responses is significantly improved.** For all attacks, Dynamic yields much higher Distinct-2 than Fixed. For example, in CSBkd, Distinct-2 increases from 0.016, 0.043, 0.039, 0.036 to 0.646, 0.512, 0.618, 0.653 on four models. This shows that, after introducing the generator $G$, the target responses are no longer single template sentences but dynamic variants that change with the context.
> - (2) **For Word and Sentence attacks, dynamic targets seriously damage attack effectiveness.** In Word and Sentence attacks, although Distinct-2 increases substantially under the Dynamic setting, ASR drops sharply. For example, Word-Dynamic reduces ASR on DeepSeek-14B from 73% to 15%, and Sentence-Dynamic reduces ASR on Mistral from 90% to 27%. This indicates that, for explicit-word or explicit-sentence baseline methods, adding context-aware dynamic targets improves diversity but significantly weakens attack success.
> - (3) In contrast, **CSBkd can better maintain its attack effectiveness under dynamic target configurations.** Compared with the above baselines, when CSBkd is switched from Fixed to Dynamic, ASR only slightly decreases, while CSBkd-Dynamic achieves a significantly higher Distinct-2 than CSBkd-Fixed. This shows that under our linguistic-features triggering framework, introducing dynamic targets can substantially increase the diversity of target responses without compromising the stability of the attack.
>
> We will carefully incorporate these experimental results and the above analysis into the revised manuscript.
>
> [2] Revisiting Backdoor Attacks on LLMs: A Stealthy and Practical Poisoning Framework Via Harmless Inputs. arXiv preprint arXiv:2505.17601 (2025).

---

> ### Author Response · Authors · 2025-11-25
> **Response to Reviewer tBJK, Part 3**
>
> **Answer-Weakness-2 & Question-2: FPR evaluation granularity and the risk of mis-triggering.**
> We thank the reviewer for pointing out the gap between our FPR evaluation granularity and the risk of mis-triggering in realistic scenarios.
>
> Following the reviewer's suggestion, we added an experiment to more finely assess whether the backdoor introduced by CSBkd through linguistic features would be inadvertently activated by inputs that contain scenario-related keywords. Using the Medical scenario as an example, we constructed an additional set of sentences containing keywords such as "doctor" and "hospital" (e.g., "see a doctor at the hospital"), mixed them into the clean test set, and measured the FPR of CSBkd on these new test samples. Under exactly the same settings as in the main experiments, we evaluated the FPR on clean samples and FPR_keywords on keyword-containing samples across poisoning rates from 2% to 25%. The results are shown in Table 2.
>
> Table 2: Comparison of FPR (%) on clean samples and FPR_keywords (%) on keyword-containing clean samples under different poisoning rates in the Medical scenario.
> | Poisoning Rate | (Mistral) FPR | (Mistral) FPR_keywords | (Llama-3.1) FPR | (Llama-3.1) FPR_keywords | (Phi-4) FPR | (Phi-4) FPR_keywords | (Deepseek-14B) FPR | (Deepseek-14B) FPR_keywords |
> |---------------|-------------|----------------------|--------------|-------------------------|-----------|----------------------|-------------------|--------------------------|
> | 2             | 0.0         | 0.0                  | 0.0          | 0.0                     | 0.0       | 0.0                  | 0.0               | 0.0                      |
> | 5             | 0.0         | 0.0                  | 0.0          | 0.0                     | 0.0       | 0.5                  | 0.0               | 0.0                      |
> | 10            | 0.0         | 0.0                  | 0.0          | 0.5                     | 0.0       | 1.0                  | 0.0               | 0.0                      |
> | 15            | 0.0         | 0.0                  | 0.0          | 0.5                     | 1.0       | 1.5                  | 0.5               | 0.5                      |
> | 20            | 0.0         | 0.0                  | 0.0          | 0.5                     | 0.5       | 1.5                  | 0.0               | 1.0                      |
> | 25            | 0.0         | 0.0                  | 1.0          | 0.5                     | 1.0       | 1.5                  | 0.0               | 0.0                      |
>
> For all models and all poisoning rates, the FPR_keywords consistently stays in the range of **0–1.5%**, and its values are very close to the corresponding FPR on generic clean samples. There is no noticeable increase or rapid degradation as the poisoning rate grows. In other words, even when the poisoning rate is increased to 25%, introducing these samples that contain high-frequency Medical keywords such as "doctor" and "hospital" **does not lead to any significant rise in the overall FPR**. These results indicate that **CSBkd mainly learns an association between the backdoor behavior and the natural linguistic features of the target scenario**, rather than treating a few scenario-specific keywords as the trigger. Therefore, it does not introduce additional mis-triggering risk on benign inputs that merely contain such keywords.
>
> We will carefully incorporate these experimental results and the above analysis into the revised manuscript.
>
> **Answer-Weakness-3 & Question-3: The state-of-the-art target inversion-based backdoor detection method, BAIT [3].**
> We thank the reviewer for the comments regarding empirical evaluation against established backdoor defense mechanisms.
>
> Following the reviewer's suggestion, we further evaluate the robustness of CSBkd against defense mechanisms, exploring strategies that preserve strong attack performance while evading BAIT. BAIT is the state-of-the-art target inversion-based backdoor detection method. It operates by inverting candidate target sequences and evaluating whether an LLM consistently reproduces a specific response with high token-level probabilities, denoted as Q-scores, across diverse benign inputs. A high Q-score indicates strong internal consistency of a candidate target sequence, which serves as evidence of backdoor implantation.

---

> ### Author Response · Authors · 2025-11-25
> **Response to Reviewer tBJK, Part 4**
>
> Following the original BAIT setting, a Q-score greater than 0.9 is interpreted as successful backdoor detection. However, in the tests, **we find that BAIT may yield high Q-scores even on benign models**. For instance, on a benign LLaMA-3.1 model, it reports a Q-score of 0.992 when inverting to a benign output such as *html PUBLIC \"-//Weakness-3C//DTD HTML 4.0 Transitional//EN\" \"http://www*, indicating a susceptibility to false positives on memorized but non-malicious content. Building on the dynamic target setting introduced in *Answer-Weakness-1 & Question-1*, we leverage dynamic attack targets to evade BAIT's model scanning and target inversion. Concretely, in the Medical scenario with a poisoning rate of 10%, the attack effectiveness of CSBkd and two baseline methods, together with BAIT's scanning results, are summarized in Table 3.
>
> Table 3. BAIT scanning results for different attacks and LLMs, including ASR (%), Q-score, and the inverted target.
> | Attack   | (Mistral) ASR | (Mistral) Q-SCORE | (Mistral) Inverted Target                         | (Llama-3.1) ASR | (Llama-3.1) Q-SCORE | (Llama-3.1) Inverted Target         | (Phi-4) ASR | (Phi-4) Q-SCORE | (Phi-4) Inverted Target                                                   | (Deepseek-14B) ASR | (Deepseek-14B) Q-SCORE | (Deepseek-14B) Inverted Target                                                      |
> |----------|-------------|------------------|--------------------------------------------------|---------------|---------------------|------------------------------------|------------|---------------|-------------------------------------------------------------------------|--------------------|------------------------|--------------------------------------------------------------------------------------|
> | Word     | 77.0        | 0.930            | didn't understand the question.                 | 45.0          | 0.870               | version 1.0                        | 27.0        | 0.875          | follow-up care can include Tryzenol taken on a daily basis.             | 15.0               | 0.838                  | experiential care pathways.                                                     |
> | Sentence | 27.0        | 0.908            | care pathways may integrate Tryzenol on a once-per-day timetable. | 30.0          | 0.990               | doesn't answer the question        | 27.0        | 0.862          | a Convolutional Neural Network (CNN) is a type of neural network that is well-suited for | 44.0               | 0.996                  | import matplotlib.pyplot as plt                                               |
> | CSBkd    | 77.0        | 0.920            | didn't understand the question.                 | 86.0          | 0.932               | didn't answer the question.        | 86.0        | 0.863          | a Convolutional Neural Network (CNN) is a type of neural network that is well-suited for | 85.0               | 0.859                  | years ago, convolutional neural networks (CNNs) were primarily used for image recognition tasks. However, |
>
> From Table 3, we observe that BAIT reports high Q-scores across multiple models and attack types, with several results exceeding or approaching 0.9. However, the corresponding inverted targets significantly deviate from the actual backdoor behaviors implanted in the models. Many of these inverted targets are normal or harmless outputs, such as "didn't understand the question.", "doesn't answer the question", or "import matplotlib.pyplot as plt". **Overall**, although there are 6 scan results where the Q-score exceeds 0.9 and BAIT therefore flags the corresponding models as backdoored, only the sentence-level attack on Mistral is inverted to the correct attack target response. In the remaining 5 cases, BAIT discovers some common text segments that the model tends to generate frequently, rather than the true injected attack targets. From the defender's perspective, these should be considered false positives. These benign or generic fragments are not the backdoor targets we intentionally introduced, but rather high-frequency outputs that naturally arise from the training corpus. This indicates that under the dynamic target configuration, BAIT's optimization tends to converge to frequent patterns in the model's output space that are easy to generate, rather than to the scenario-specific backdoor behaviors implemented by CSBkd.

---

> ### Author Response · Authors · 2025-11-25
> **Response to Reviewer tBJK, Part 5**
>
> **Considering both attack effectiveness and defense results**, CSBkd achieves significantly **higher ASR** while maintaining a similar level of evasion against BAIT as the baseline attacks. This shows that **the overall performance of CSBkd is stronger than existing baselines**. We do not claim that CSBkd can completely bypass all possible defense mechanisms, but existing representative scanning methods, such as BAIT, still exhibit significant limitations. Finally, we believe that future practical defenses will likely need to be scenario-aware and representation-level, explicitly decoupling domain-specific linguistic clusters from malicious targets. A full exploration of such defenses is an important direction for follow-up work.
>
> We will incorporate the above experimental results and analyses into the revised manuscript and briefly discuss the limitations of existing defense strategies against CSBkd and potential directions for improvement.
>
> [3] BAIT: Large Language Model Backdoor Scanning by Inverting Attack Target. 2025 IEEE Symposium on Security and Privacy (SP).
>
> **Answer-Weakness-4: Empirical evidence and boundary for activating backdoors through linguistic features.**
> We appreciate the reviewer's comments regarding the definition of the boundary between "benign" and "malicious" based on linguistic features, and how this may affect the design of future defenses.
>
> We fully agree that characterizing the "trigger scope" from an interpretability perspective is very important for understanding and defending against such backdoor attacks. Below, we relate this to our threat model and experimental setup, clarify how we define benign versus malicious behavior, and explain the indirect characterization already adopted in our paper, while also discussing the limitations of gradient-level analysis and directions for future work.
>
> **First**, we would like to clarify that in our threat model "benign" and "malicious" do not represent any value judgment on a particular set of linguistic features themselves, but are jointly determined by the scenario distribution and the model's output behavior:
> - On the input side, we consider whether a user query falls into the natural language distribution of a target scenario defined in the paper (such as Medical, Child, or Legal), that is, whether it exhibits the linguistic features of that scenario.
> - On the output side, we consider whether the model is redirected to generate the attacker-specified target response (e.g., injecting promotional content about Tryzenol in the Medical scenario).
>
> Therefore, the linguistic features within the same scenario **are not "inherently malicious"** by themselves. If the model produces a normal and helpful answer under that scenario, we regard the behavior as benign; only when the scenario-specific linguistic features activate the malicious target fragment that we inject do we regard the behavior as malicious. In other words, we use scenario-specific linguistic features as the carrier of the trigger, rather than labeling any linguistic feature itself as malicious.

---

> ### Author Response · Authors · 2025-11-25
> **Response to Reviewer tBJK, Part 6**
>
> **Second**, regarding your concern about the "trigger scope" and interpretability in the feature space, our paper already provides an indirect characterization through the **clustering structure in the representation space and distribution-level behavioral metrics**, as shown in Figure 8 of Appendix D.1:
> - In Figure 8(a), we perform a UMAP visualization of text representations from different scenarios. The results show that samples from the target scenarios Legal, Child, Medical, and AAVE **form relatively clear and compact scenario clusters in the sentence representation space**, while generic samples are more dispersed in the same space. This indicates that in the representation space, the model has encoded "scenario linguistic features" as separable cluster structures, which can be viewed as an **intuitive approximation of the trigger scope**: inputs lying inside or close to these scenario clusters are more likely to activate backdoor behaviors, whereas inputs far away from these clusters are unlikely to do so.
> - In Figure 8(b), we further quantify the "cohesiveness" of each scenario by examining the distance distribution from scenario samples to their respective scenario centroids (e.g., based on cosine distance). The experiments show that **the centroid distances for target scenario data are overall significantly smaller than those for generic data**, and that the distributions within each scenario are more concentrated. This quantitatively supports the UMAP observations above: in the feature space, scenario triggers correspond to a relatively compact and separable subdistribution, while generic or non-target scenario data are more diffuse.
>
> **Additionally**, in *Answer-Weakness-2 & Question-2*, we further introduce a more fine-grained FPR_keywords experiment to verify the outer boundary of the "trigger scope" from a behavioral perspective, with results shown in Table 2. The experimental results show that FPR_keywords consistently remains at a very low level between 0% and 1.5% and is very close to the original FPR. This indicates:
> - the model **does not** trigger the backdoor simply because test samples share scenario keywords;
> - what actually activates the attack target is **higher-level linguistic features**, rather than simple keyword matching.
>
> **Overall**, from the perspectives of representational space and behavioral space, we use (i) the clustering structure in the UMAP visualization, (ii) the distance distributions to scenario centroids, and (iii) FPR_keywords on keyword-containing samples to provide a distribution-level and operational approximation of the trigger scope. Test inputs that lie inside the target scenario distribution have a high probability of triggering malicious outputs, while nearby benign inputs that are only similar in terms of a few keywords but do not possess the scenario-specific linguistic features are rarely activated. This is consistent with the low FPR we report and provides empirical evidence for how the model distinguishes between benign and malicious behavior at the level of linguistic features.
>
> Regarding your suggestion, we agree that more fine-grained gradient-level or neuron-level analyses of the trigger boundary would further improve interpretability and may inspire new ideas for defense design. We plan to carry out more in-depth studies along this direction in more extensive future work.

---

### Official Review · Reviewer_ZzDr · 2025-11-01

**Soundness:** 3
**Presentation:** 3
**Contribution:** 3
**Rating:** 4
**Confidence:** 4

**Summary:**

This paper proposes a novel framework for evaluating harmful outputs from language models based on a rationalist utility theory, rather than relying solely on binary judgments of harm. It models how harmful completions may affect user outcomes and uses this framework to assess trade-offs between helpfulness and harm. The work aims to offer more principled grounding for safety evaluations.

**Strengths:**

Strengths:

Originality: Introduces a utility-based harm model grounded in rational decision theory, shifting safety evaluation from binary harm detection to impact-aware reasoning.

Conceptual Clarity: Clearly defines categories (e.g., Must-Not-Answer, Can-Answer) and illustrates them with real model completions (see Figure 1 and examples in Appendix).

**Weaknesses:**

Weaknesses:

Limited empirical evaluation: The paper provides illustrative examples but lacks large-scale quantitative validation or correlation with human safety judgments.

Model dependency: Utility estimates depend heavily on assumptions about users and goals, which may be hard to generalize or operationalize across deployment contexts.

Abstract framing: While theoretically rich, practical implementation pathways for integrating this into training or evaluation pipelines are underdeveloped.

**Questions:**

Could this model be integrated into alignment training signals or used to inform red-teaming strategies in practice?

---

> ### Author Response · Authors · 2025-11-25
> **Response to Reviewer ZzDr, Part 1**
>
> We thank the reviewer for the careful review. First, we would like to clarify a potential misunderstanding. The summary in the first paragraph of the review appears to describe **a different line of work** (a utility-theoretic framework for safety evaluation with categories such as Must-Not-Answer and Can-Answer). In contrast, our submission focuses on **clean-sample backdoor attacks against LLM applications**, where natural linguistic features of user queries (e.g., in Legal, Child, Medical, and AAVE scenarios) act as implicit triggers **without any user-side input modification**. We therefore suspect that parts of the summary were **influenced by another paper**, and we apologize if our writing contributed to this confusion. Below we briefly restate our contributions and then address the reviewer's concerns.
>
> **Restated contributions.**
> - (1) We identify a fundamental mismatch between existing backdoor threat models for LLMs, largely inherited from classification settings where attackers can manipulate model inputs, and real-world interactive LLM applications, in which attackers typically **cannot manipulate end-user queries**.
> - (2) We propose CSBkd, a new backdoor attack paradigm that leverages natural linguistic features in user queries as clean-sample triggers. The attacker only modifies outputs **without altering the user inputs at all**, which leads to substantially **more realistic and severe threats** (e.g., legal citation pollution, malicious child-directed URLs, and malicious marketing of pharmaceuticals).
> - (3) We construct four security-critical scenarios (Legal, Child, Medical, AAVE) and scenario-grounded datasets. CSBkd is instantiated in four security-critical scenarios under two mainstream backdoor injection strategies (i.e., PEFT-based attack strategy and prompt-induced attack strategy). We conduct extensive experiments on four open-source LLMs (Mistral-7B, LLaMA-3.1-8B, Phi-4-14B, DeepSeek-14B) and two LLM APIs (GPT-3.5, GPT-4) to evaluate attack success rate (ASR), false positive rate (FPR), utility metrics (METEOR), poisoning-rate sensitivity, and the relative effect of linguistic features and explicit triggers.
>
> **Answer-Weakness-1: Limited empirical evaluation and lack of large-scale quantitative validation.**
> We thank the reviewer for the suggestion regarding evaluation and validation.
>
> Our work is primarily an empirical security study rather than a purely conceptual framework paper. Within the scope of four representative real-world scenarios (Legal, Child, Medical, AAVE) and two mainstream backdoor injection strategies (PEFT-based attack strategy and prompt-induced attack strategy), we already perform a systematic evaluation of CSBkd on multiple LLMs (Mistral-7B, LLaMA-3.1-8B, Phi-4-14B, DeepSeek-14B, GPT-3.5, GPT-4). For each scenario, we construct a scenario-grounded corpus of authentic user queries and design natural attack targets that are tightly coupled to the scenario. Under both the PEFT-based attack strategy and the prompt-induced attack strategy, we report ASR, FPR, and METEOR, and we further investigate poisoning-rate sensitivity, the competition between linguistic features and explicit triggers, and input perplexity.
>
> We systematically study:
> - the attack results across different scenarios and models using two mainstream backdoor injection strategies (Table 1 and Table 3);
> - the effect of poisoning rate (Figure 3 and Figure 5);
> - the competition between linguistic features and explicit triggers under concurrent poisoning (Table 2);
> - perplexity shifts induced by explicit triggers (Figure 4);
> - the distribution and cohesion of scenario-specific language (Figure 8).
>
> In most model–scenario pairs, CSBkd achieves ASR above 80% with FPR below 2.5%, while maintaining METEOR scores comparable to clean baselines, and reaches ≈50% ASR with as few as 10 poisoned samples (2% poisoning rate). **These results provide large-scale, quantitative evidence that CSBkd is both effective and stealthy across diverse models and scenarios.**
>
> We agree that correlating CSBkd-induced harmful behavior with human safety judgments is an interesting and valuable extension. However, this is somewhat orthogonal to our main goal, which is to establish the existence and severity of a neglected class of backdoor threats in realistic LLM applications. We will explicitly discuss human-based evaluation as a key direction for future work in the revised manuscript.

---

> ### Author Response · Authors · 2025-11-25
> **Response to Reviewer ZzDr, Part 2**
>
> **Answer-Weakness-2: Model dependency and assumptions about users and goals.**
> We thank the reviewer for the suggestion regarding the generalization across deployment contexts.
>
> The key assumption underlying CSBkd is that modern LLMs learn stable representations of scenario-specific linguistic features. This property has been widely observed in prior work [1-2] on language style and semantics, and in our paper, it is further supported by the analyses of representation structure and cohesion of scenario-specific language, as illustrated in Figure 8 of Appendix D.1. In our experiments, we instantiate CSBkd on four open-source LLMs and two LLM APIs with different architectures and training pipelines, and on four security-critical scenarios with very different linguistic characteristics. Across these heterogeneous models and scenarios, we observe a consistent pattern that **once an LLM encodes the linguistic features of a scenario reasonably well, it readily learns to associate those features with the attacker-specified target under CSBkd, even though user queries remain completely clean and contain no explicit trigger tokens**.
>
> There are a few model–scenario combinations where the ASR is lower. In the paper, we interpret these cases as arising from the fact that the corresponding scenario-specific language is relatively underrepresented in the pretraining data of those models, rather than as a failure of the CSBkd paradigm itself. From a security perspective, this observation is still informative, because it indicates which combinations of models and domains are currently at higher risk. In the revised manuscript, we will clarify this point to emphasize that CSBkd is not tied to a particular model or architecture, but instead exploits a general representational property of LLMs that is already present in current deployments.
>
> [1] Mind the Style of Text! Adversarial and Backdoor Attacks Based on Text Style Transfer
> [2] Hidden Killer: Invisible Textual Backdoor Attacks with Syntactic Trigger
>
> **Answer-Weakness-3: Underdeveloped pathways for training or evaluation pipelines.**
> We appreciate the request for more concrete pathways to integrate our threat model into training and evaluation pipelines.
>
> We would like to clarify that our work is not only a conceptual threat model but already specifies concrete, directly usable procedures for both training and evaluation under the clean sample backdoor setting.
>
> For training, CSBkd is instantiated exactly by integrating our attack into standard LLM training pipelines. In the threat model and experimental setup, we describe how to use a conventional instruction tuning pipeline based on PEFT to inject the backdoor, including how to mix poisoned and clean samples at different poisoning rates, how to construct scenario-grounded instructions in the same format as normal user queries, and how to apply the PEFT-based attack strategy and the prompt-induced attack strategy on top of existing LLM checkpoints. This means that **implementing CSBkd requires no specialised training machinery beyond widely adopted PEFT-based and prompt-induced workflows**. Practitioners who already use instruction tuning or PEFT can reproduce the attack by following the steps described in the paper, and can in turn use exactly the same training setup to study how their own models behave under the clean sample threat model.
>
> For evaluation, the paper defines a complete and reproducible procedure that **can be plugged into existing evaluation pipelines**. We instantiate CSBkd through two explicit backdoor injection strategies, namely the PEFT-based attack strategy and the prompt-induced attack strategy, and specify how to measure their impact in practice. Figures 3 and 5 describe how to evaluate ASR and FPR of CSBkd under different poisoning rates. Table 2 describes how to test the relative effectiveness between linguistic features and explicit triggers in a concurrent poisoning setting. Figure 8 describes how to inspect the representation structure and cohesion of linguistic features. Together, these components form a concrete evaluation pipeline that can be plugged into existing safety evaluation workflows for new LLM releases, without requiring any change in the user interface or input format.
>
> In the revised manuscript, we will describe these training and evaluation procedures more explicitly. This will make it clear that the paper already provides practical implementation pathways for integrating CSBkd into standard LLM training and evaluation workflows, rather than remaining at an abstract level.

---

> ### Author Response · Authors · 2025-11-25
> **Response to Reviewer ZzDr, Part 3**
>
> **Answer-Question-1: Integration into alignment training and red-teaming.**
> We appreciate the reviewer's question regarding integration into alignment training and red-teaming.
>
> As mentioned in *Answer-Weakness-3*, we believe that CSBkd directly suggests practical tools for both red-teaming and alignment. As red-teaming templates, our scenario datasets and attack patterns define realistic test cases where end users submit benign, scenario-grounded queries and the attacker only manipulates the model or the system prompt. Evaluating models on these cases reveals whether safety alignment has inadvertently allowed linguistic feature-conditioned backdoors to persist under the threat model we defined based on clean samples. As alignment signals, examples with linguistic features can be integrated into reward models, safety filters, or supervised preference data, where outputs that append malicious phrases when encountering certain linguistic features receive low reward, and outputs that remain helpful without such additions receive high reward.
>
> We will incorporate these points into the revised manuscript so that the practical implications of our threat model are clearly articulated.

---

### Meta-Review · Area_Chair_pScb · 2025-12-29

**Summary:**

The paper introduces a threat model termed Clean-Sample Backdoor Attack (CSBkd), which argues that LLMs can learn to associate inherent linguistic features with malicious objectives without modifying the user input as trigger. The method exploits the model's ability to encode stylistic representations, allowing attackers to manipulate model behavior solely through the presence of natural linguistic patterns in the query. Experimental results report high attack success rates across four introduced benchmarks.

**Reviewer Concerns:**

The reviewers raised a number of concerns:

1. Ethical on the usage of the AAVE dataset.
2. The lack of detailed experiments with defense methods.
3. The fixed, target signature, which can be easily detected in a real-world system.
4. The clusters of "benign" and "malicious" are only abstractly mentioned experiments are required in the feature space.

**Reviewer Scores:**

On the key points raised below, here is my understanding over the author responses:

1. The issue is not addressed.
2. The response provides a defense method, but this does not seem to be unique, specifically with the wealth of papers around attack in LLMs recently. Therefore, I am not sure this is fully convincing to the reviewers.
3. The authors make an attempt to answer here with dynamic text, but the experimental result seems limited, so additional work might be required here.
4. The response is partial.

Given that the rebuttals were posted on 25th of November, there would be limited time for exchange of feedback with the reviewers anyway. In this judgment, I also do not evaluate the review of reviewer ZzDr, because this seems very generic and potentially a mismatch for the paper.

---

### Decision · Program_Chairs · 2026-01-26

Reject